# Interactive Parallel Exploration for Reinforcement Learning in Continuous Action Spaces

## Abstract

In this paper, a new interactive parallel learning scheme is proposed to enhance the performance of off-policy continuous-action reinforcement learning. In the proposed interactive parallel learning scheme, multiple identical learners with their own value-functions and policies share a common experience replay buffer, and search a good policy in collaboration with the guidance of the best policy information. The information of the best policy is fused in a soft manner by constructing an augmented loss function for policy update to enlarge the overall search space by the multiple learners. The guidance by the previous best policy and the enlarged search space by the proposed interactive parallel learning scheme enable faster and better policy search in the policy parameter space. Working algorithms are constructed by applying the proposed interactive parallel learning scheme to several off-policy reinforcement learning algorithms such as the twin delayed deep deterministic (TD3) policy gradient algorithm and the soft actor-critic (SAC) algorithm, and numerical results show that the constructed IPE-enhanced algorithms outperform most of the current state-of-the-art reinforcement learning algorithms for continuous action control.

## 1 Introduction

Reinforcement learning (RL) for continuous action control is an active research field. In RL, an agent learns a policy through interaction with the environment to maximize the cumulative reward. One of the key issues in RL is the trade-off between exploitation and exploration. Exploitation is to make a best decision based on the already collected information, whereas exploration is to collect more new information about the environment. The balance between the two is important for good RL algorithms. For example, DQN (Mnih et al. (2015)) balances exploitation and exploration by taking actions based on the $\epsilon$-greedy approach. Deep deterministic policy gradient (DDPG) (Lillicrap et al. (2015)) and twin delayed deep deterministic (TD3) (Fujimoto et al. (2018)) policy gradient algorithms promote exploration by adding Ornstein-Uhlenbeck noise and Gaussian noise to the best decision action, respectively. Soft actor-critic (SAC) (Haarnoja et al. (2018)) performs balancing by using a maximum entropy objective. However, most of the previous works focus on exploration to obtain unobserved states or actions. In this paper, we consider exploration in the policy parameter space by using parallel identical learners for the same environment. By having multiple identical learners for the same environment, we can have increased search capability for a better policy. Parallelism in learning has been investigated widely in distributed RL (Nair et al. (2015), Mnih et al. (2016), Horgan et al. (2018), Barth-Maron et al. (2018), Espeholt et al. (2018)), evolutional strategies (Salimans et al. (2017), Choromanski et al. (2018)), and recently in population based training (PBT) (Jaderberg et al. (2017), Jaderberg et al. (2018), Conti et al. (2017)) for faster and better search for parameters and/or hyperparameters. In this paper, we also apply parallelism to RL in order to enhance the learning performance but in a slightly different way as compared to the previous methods.

The proposed algorithm is intended for any off-policy RL algorithms and is composed of a chief, $N$ environment copies of the same environment, and $N$ identical learners with a shared common experience replay buffer and a common base algorithm, as shown in Fig. 1. Each learner has its own value function(s) and policy, and trains its own policy by interacting with its own environment

copy with some additional interaction with the chief, as shown in Fig. 1. At each time step, each learner takes an action to its environment copy by using its own policy, stores its experience to the shared common experience replay buffer. Then, each learner updates its value function parameter and policy parameter by drawing mini-batches from the shared common replay buffer by minimizing its own value loss function and policy loss function, respectively.

One way to implement parallel learning under the above setup is to run $N$ fully independent parallel learning without interaction among the learners except sharing their experiences until the end of time steps and to choose the policy from the learner with the maximum accumulated reward at the end for future use. We will refer to this method as the experience-sharing-only method. However, this method ignores possible benefit from mutual interaction among the learners during training. In order to harness the benefit of mutual interaction among the learners in parallel learning, we exploit the information from the best learner among all learners periodically during training like in PBT (Jaderberg et al. (2017), Jaderberg et al. (2018)). Suppose that the value and policy parameters of each learner are initialized and learning is performed as described above for $M$ time steps. At the end of $M$ time steps, we can determine who is the best learner based on the average of the most recent $E_r$ episodic rewards for each learner. Then, the policy parameter information of the best learner can be used to enhance the learning of other learners for the next $M$ time steps. This information can help learners stuck in local minima escape from the local minima and guide other learners for better direction.

One simple way to exploit this best policy parameter information is that we reset the policy parameter of each learner with the policy parameter of the best learner at the beginning of the next $M$ time steps, make each learner perform learning from this initial point in the policy parameter space for the next $M$ time steps, select the best learner again at the end of the next $M$ time steps, and repeat this procedure every $M$ time steps in a similar way that PBT (Jaderberg et al. (2017)) copies the best learner's parameters and hyperparameters to other learners. We will refer to this method as the reloading method in this paper. However, this reloading method has the problem that the search area covered by all $N$ searching policies collapses to one point at the time of parameter copying and thus the search area can be narrow around the previous best policy point. In order to overcome such disadvantage, instead of resetting the policy parameter with the best policy parameter every $M$ time steps, we here propose using the policy parameter information of the best learner in a soft manner to enhance the performance of the overall parallel learning. In the proposed scheme, the shared best policy information is used only to guide the policies of other learners. The policy of each learner is updated by improving the performance around a certain distance from the shared guiding policy. The chief periodically determines the best policy among the policies of all learners and distributes the best policy parameter to all learners so that the learners search for better policies around the previous best policy. The chief also enforces that the $N$ searching policies are spread in the policy parameter space with a given distance from the previous best policy point so that the search area in the policy space by all $N$ learners maintains a wide area and does not collapse into a narrow region.

The proposed interactive parallel exploration (IPE) learning method can be applied to any off-policy RL algorithms and implementation is easy. Furthermore, the proposed method can be extended directly to distributed or multi-agent RL systems. We apply our IPE scheme to the TD3 algorithm and the SAC algorithm, which are state-of-the-art off-policy algorithms, as our base algorithms, and the new algorithms are named IPE-TD3 and IPE-SAC algorithms, respectively. Numerical result shows that the IPE-enhanced algorithms outperform the baseline algorithms both in the speed of convergence and in the final steady-state performance. The gain by IPE

## 2  BACKGROUND AND RELATED WORKS

### 2.1  DISTRIBUTED RL

Distributed RL is an efficient way that takes advantage of parallelism to achieve fast training for large complex tasks (Nair et al. (2015)). Most of the works in distributed RL assume a common structure composed of multiple actors interacting with multiple copies of the same environment and a central system which stores and optimizes the common Q-function parameter or the policy parameter shared by all actors. The focus of distributed RL is to optimize the Q-function parameter or the policy parameter fast by generating more samples for the same wall clock time with multiple actors. In order to achieve this goal, researchers investigated various techniques for distributed RL, e.g.,

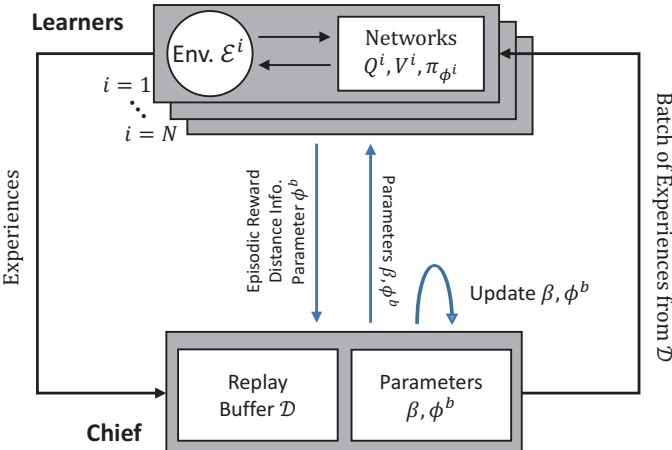

Figure 1: The overall structure of the proposed IPE scheme

asynchronous update of parameters (Mnih et al. (2016), Babaeizadeh et al. (2017)), sharing an experience replay buffer (Horgan et al. (2018)), GPU-based parallel computation (Babaeizadeh et al. (2017), Clemente et al. (2017)), GPU-based simulation (Liang et al. (2018)) and V-trace in case of on-policy algorithms (Espeholt et al. (2018)). Distributed RL yields significant performance improvement in terms of the wall clock time but it does not consider the possible enhancement by interaction among multiple learners like in IPE and PBT. The proposed IPE uses a similar structure to that in (Nair et al. (2015), Espeholt et al. (2018)): that is, IPE is composed of multiple learners and a central system called chief. The difference is that each learner has its own Q or value function parameter and policy parameter and optimizes the parameters in parallel with interactions.

## 2.2 POPULATION BASED TRAINING

Parallelism is also exploited in finding optimal parameters and hyperparameters of training algorithms for neural networks in PBT (Jaderberg et al. (2017), Jaderberg et al. (2018), Conti et al. (2017)). PBT (Jaderberg et al. (2017)) first chooses multiple sets of hyperparameters and parameters for a common base algorithm, and runs the base algorithm separatively in parallel at multiple learners to train their neural networks using those parameters and hyperparameters. Each learner updates the neural network parameters by perturbing the assigned hyperparameters. During training, in principle PBT evaluates the performance of multiple learners periodically, and selects the best performing hyperparameters, and then distributes the best performing hyperparameters and the corresponding parameters to other learners, although implementation details can be changed. Recently, PBT is applied to competitive multi-agent RL (Jaderberg et al. (2018)) and novelty search algorithms (Conti et al. (2017)).

Although PBT is mainly developed to tune hyperparamters, the philosophy of PBT can be applied to find optimal parameters for given hyperparameters by multiple learners. In this case, multiple learners update their parameters in parallel, their performance is measured periodically, the parameters of the best performing learner are copied to other learners, other learners independently update their parameters from the copied parameters as their new initialization, and this process is repeated. The proposed IPE is similar to PBT in the sense that it exploits the parameters of the best performing learner among multiple parallel learners. However, IPE is different from the PBT-derived method in the way how IPE uses the parameters of the best learner. In the PBT-derived method, the parameters of the best learner are copied to other learners and other learners' parameters are reset to the parameters of the best performing learner. Then, the parameters of each learner are updated by stochastic gradient descent (SGD). However, in IPE the parameters of the best performing learner are not copied but used in a soft manner as a guiding direction. Copying means that the parameters of all learners collapse to a single point in the parameter space. Furthermore, unlike PBT, IPE uses a common experience replay buffer to store all experiences from multiple learners with different parameters to exploit the diverse experiences of multiple learners with different parameters.

As mentioned in Section 1, we refer to the PBT-derived method with a common experience replay buffer as the reloading method of which performance will be given in ablation study in Section 4.

Although IPE is considered only for parallel parameter search in this paper, combining the soft way of using the parameters of the best performing learner with hyperparameter search is an interesting future work.

## 2.3 GUIDED POLICY SEARCH

Our IPE method is also related to guided policy search (Levine & Koltun (2013), Levine et al. (2016), Teh et al. (2017), Ghosh et al. (2018)). Recently, Teh et al. (2017) proposed a guided policy search method for joint training of multiple tasks in which a common policy is used to guide local policies and the common policy is distilled from the local policies. Here, the local policies' parameters are updated to maximize the performance and minimize the KL divergence between the local policies and the common distilled policy. The proposed IPE is related to guided policy search in the sense that multiple policies are guided by a common policy. However, the difference is that the goal of IPE is not learning multiple tasks but learning optimal parameters for a common task as in PBT and hence the guiding policy is not distilled from multiple local policies but chosen as the best performing policy among multiple learners.

## 2.4 EXPLORATION

Improving exploration has been one of the key issues in RL and many different ways have been developed to improve exploration through maximum entropy objectives (Haarnoja et al. (2017; 2018)), noise in networks (Fortunato et al. (2018); Plappert et al. (2018)), and intrinsically motivated approaches (Bellemare et al. (2016); Ostrovski et al. (2017); Pathak et al. (2017); Achiam & Sastry (2017); Zheng et al. (2018)). The proposed IPE also enhances exploration. Specifically, IPE uses exploitation for exploration. Exploitation for exploration has been considered in the previous works (White & Sofge (1992), Oh et al. (2018)). In particular, Oh et al. (2018) exploited past good experiences to explore the sample space, whereas IPE exploit the current good policy among multiple policies to explore the policy space.

## 2.5 THE SET UP: PARALLEL LEARNING FOR A COMMON ENVIRONMENT

The considered parallel learning setting consists of the environment $\mathcal{E}$ and $N$ policies $\{\pi^1, \cdots, \pi^N\}$. The environment $\mathcal{E}$ is described as a Markov decision process (MDP) defined by the tuple $\langle \mathcal{S}, \mathcal{A}, \mathcal{T}, r \rangle$, where $\mathcal{S}$ is the state space, $\mathcal{A}$ is the action space, $\mathcal{T} : \mathcal{S} \times \mathcal{A} \times \mathcal{S} \rightarrow [0,1]$ is the state transition probability, and $r : \mathcal{S} \times \mathcal{A} \rightarrow \mathbb{R}$ is the reward function. There exist $N$ copies $\{\mathcal{E}^1, \cdots, \mathcal{E}^N\}$ of the environment $\mathcal{E}$, i.e., $\mathcal{E}^1 = \cdots = \mathcal{E}^N = \mathcal{E}$, and the $N$ environment copies may have different random initial seeds. The policy $\pi^i$ interacts with its corresponding environment copy $\mathcal{E}^i$ and builds up its trajectory $\{(s_t^i, a_t^i, r_t^i), t = 1, 2, \cdots\}$ for each $i = 1, \cdots, N$. At time step $t$, the environment copy $\mathcal{E}^i$ has a state $s_t^i \in \mathcal{S}$. The policy $\pi^i$ interacts with the environment copy $\mathcal{E}^i$ by taking an action $a_t^i$ according to $\pi^i$ given the current state $s_t^i$. Then, the environment copy $\mathcal{E}^i$ yields the reward $r_t^i = r(s_t^i, a_t^i)$ and makes transition to the next state $s_{t+1}^i$ according to $\mathcal{T}$.

In this paper, in order to account for the actual amount of interaction with the environment, we define *environment steps* as the total number of interactions by all $N$ parallel policies with all $N$ environment copies. Suppose that all $N$ policies generate their trajectories simultaneously in parallel, and suppose that $M$ time steps have elapsed. Then, although the number of elapsed time steps is $M$, the number of environment steps is $NM$.

## 3 INTERACTIVE PARALLEL POLICY EXPLORATION

We now present the proposed IPE scheme with the parallel environment learning setting described in Section 2.5, and the overall structure is described in Fig. 1.

We have $N$ identical parallel learners with a shared common experience replay buffer $\mathcal{D}$, and all $N$ identical learners employ a common base algorithm, which can be any off-policy RL algorithm. The execution is in parallel. The $i$-th learner has its own environment $\mathcal{E}^i$, which is a copy of the common

environment $\mathcal{E}$, and has its own value function (e.g., Q-function) parameter $\theta^i$ and policy parameter $\phi^i$. The $i$-th learner interacts with the environment copy $\mathcal{E}^i$ with some additional interaction with the chief, as shown in Fig. 1. At each time step, the $i$-th learner performs an action $a_t^i$ to its environment copy $\mathcal{E}^i$ by using its own policy $\pi_{\phi^i}$, stores its experience $(s_t^i, a_t^i, r_t^i, s_{t+1}^i)$ to the shared common experience replay buffer $\mathcal{D}$ for all $i = 1, 2, \cdots, N$. Note that one time step corresponds to $N$ environment steps. Then, at each time step, each learner updates its value function parameter and policy parameter *for N times* by drawing $N$ mini-batches of size $B$ from the shared common replay buffer $\mathcal{D}$ by minimizing its own value loss function and policy loss function, respectively. The $N$ time updates for each learner for each time step is to exploit the samples provided by other $N - 1$ learners stored in the shared common replay buffer.

In order to harness the benefit of mutual interaction among the learners in parallel learning, we exploit the information from the best learner periodically during training like in PBT (Jaderberg et al. (2017)). Suppose that the Q-function parameter and the policy parameter of each learner are initialized and learning is performed as described above for $M$ time steps. At the end of the $M$ time steps, we determine who is the best learner based on the average of the most recent $E_r$ episodic rewards for each learner. Let the index of the best learner be $b$. Then, the policy parameter information $\phi^b$ of the best learner can be used to enhance the learning of other learners for the next $M$ time steps. Here, instead of copying $\phi^b$ to other learners, we propose using the information of $\phi^b$ in a soft manner to enhance the performance of the overall parallel learning. That is, during the next $M$ time steps, whereas we set the loss function $\tilde{L}(\theta^i)$ for the Q-function to be the same as the loss $L(\theta^i)$ of the base algorithm, we set the loss function $\tilde{L}(\phi^i)$ for the policy parameter $\phi^i$ of the $i$-th learner as the following *augmented* version:

$$\tilde{L}(\phi^i) = L(\phi^i) + \mathbf{1}_{\{i \neq b\}} \beta \hat{\mathbb{E}}_{s \sim \mathcal{D}} \left[ D(\pi_{\phi^i}, \pi_{\phi^b}) \right] \tag{1}$$

where $L(\phi^i)$ is the policy loss function of the base algorithm, $\mathbf{1}_{\{\cdot\}}$ denotes the indicator function, $\beta(> 0)$ is a weighting factor, $D(\pi, \pi')$ be some distance measure between two policies $\pi$ and $\pi'$, and $\hat{\mathbb{E}}_{s \sim \mathcal{D}}$ denotes the sample expectation based on mini-batch drawn randomly from the experience replay buffer $\mathcal{D}$. The augmented loss function $\tilde{L}(\phi^i)$ in (1) is composed of two terms $L(\phi^i)$ and $\mathbf{1}_{\{i \neq b\}} \beta \hat{\mathbb{E}}_{s \sim \mathcal{D}} \left[ D(\pi_{\phi^i}, \pi_{\phi^b}) \right]$. Thus, for the non-best learners in the previous $M$ time steps, the gradient of $\tilde{L}(\phi^i)$ is the mixture of two directions: one is to maximize the return by itself and the other is to follow the previously best learner's policy. The second term in the right-hand side (RHS) of (1) guides non-best learners towards a good direction in addition to each leaner's self search.

## 3.1 DESIGN OF THE WEIGHTING FACTOR $\beta$

In (1), the weighting factor $\beta$ is common to all $N$ learners and should be determined judiciously to balance between improving its performance by each learner itself and going towards the previous best policy among the $N$ learners. We adopt an adaptive method to determine the value of $\beta$ as follows:

$$\beta = \begin{cases} \beta \leftarrow 2\beta & \text{if } \hat{D}_{best} \geq \max\{\rho \hat{D}_{change}^b, d_{search}\} \times 1.5 \\ \beta \leftarrow \beta/2 & \text{if } \hat{D}_{best} < \max\{\rho \hat{D}_{change}^b, d_{search}\}/1.5 \end{cases} \tag{2}$$

where $\hat{D}_{best}$ is the estimated distance between $\pi_{\phi^i}$ and $\pi_{\phi^b}$ averaged over all $N-1$ non-best learners, and $\hat{D}_{change}^b$ is the estimated distance between $\pi_{\phi_{updated}^b}$ (i.e., the policy of the current best learner at the end of the current $M$ time steps) and $\pi_{\phi^b}$ (i.e, the policy of the current best learner at the end of the previous $M$ time steps), given respectively by

$$\hat{D}_{best} = \frac{1}{N-1} \sum_{i \in I^{-b}} \hat{\mathbb{E}}_{s \sim \mathcal{D}} \left[ D(\pi_{\phi^i}, \pi_{\phi^b}) \right] \text{ and } \hat{D}_{change}^b = \hat{\mathbb{E}}_{s \sim \mathcal{D}} \left[ D(\pi_{\phi_{updated}^b}, \pi_{\phi^b}) \right]. \tag{3}$$

Here, $I^{-b} = \{1, \ldots, N\} \setminus \{b\}$, and $d_{search}$ and $\rho$ are predetermined hyperparameters.

This adaptation method is similar to that used in proximal policy optimization (PPO) (Schulman et al. (2017)). The update of $\beta$ is done every $M$ time steps and the updated $\beta$ is used for the next $M$ time steps. First, suppose that we do not have the first term $\rho \hat{D}_{change}^b$ in the maximum of the condition in (2). Then, when the estimated average distance $\hat{D}_{best}$ from the best policy to

the remaining policies is smaller than $d_{search}/1.5$, the parameter $\beta$ is decreased by half. Hence, the movement in the gradient direction of the second term in the RHS of (1) is diminished and the independent movement into the optimization direction for $L(\phi^i)$ becomes more important. So, each policy gradually diverges from the previous best policy serving as the reference point due to internal exploration mechanism such as added noise in action. On the other hand, when $\hat{D}_{best}$ is larger than $1.5d_{search}$, the parameter $\beta$ increases by factor two and the movement towards the previous best policy becomes more important. As time steps elapse, $\beta$ is settled down so that $\hat{D}_{best}$ is around $d_{search}$. Hence, the proposed IPE scheme searches a wide area with rough radius $d_{search}$ around the best policy in the policy parameter space, as shown in Fig. 2(a). Furthermore, with the first term $\rho\hat{D}^b_{change}$ in the maximum of the condition in (2), we can control the speed of tracking the best policy. $\hat{D}^b_{change}$ measures the speed of change in the best policy parameter. When the best policy parameter change scaled by $\rho$, i.e., $\rho\hat{D}^b_{change}$ is less than $d_{search}$, the term is invisible by the maximum operation in (2). On the other hand, when $\rho\hat{D}^b_{change} > d_{search}$, the term is active and it means that the best policy parameter changes fast. Thus, the tracking speed should be controlled. If $\hat{D}_{best} > \rho\hat{D}^b_{change}$, i.e., the distance from $\pi_{\phi^i}$ to $\pi_{\phi^b}$ is larger than $\rho$ times the distance from $\pi_{\phi^b_{updated}}$ to $\pi_{\phi^b}$, then this means that the speed of tracking the best policy is slow. Hence, we increase $\beta$ by factor two. Otherwise, we decrease $\beta$ by half. When the search for the current $M$ time steps is finished, the new best learner is selected and a new search for a wide area around the new best learner's policy $\pi_{\phi^b}$ is performed, as illustrated in Fig. 2(b). The policy parameter information $\phi^b$ of the best learner can be changed before the next best learner selection.

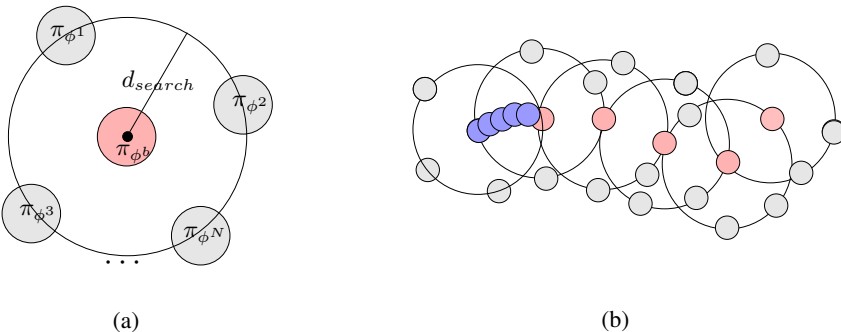

(a)  (b)

Figure 2: (a) the conceptual search coverage in the policy parameter space by parallel learners (the proper individual search area by each learner may be larger than that in the figure) and (b) an illustration of search comparison: single search (blue) versus the proposed interactive parallel search guided by the policy of best learner at each search interval (pink - policies of best learners during search)

Now, the overall procedure for the proposed IPE scheme is explained with the diagram in Fig. 1. The value function parameter and policy parameter of each learner are initialized. The chief distributes the parameter $\beta$ and the reference policy parameter $\phi^b$, which is the policy information of the best learner over the previous $M$ time steps, to all learners. At each time step, each learner interacts with its own environment copy by taking its action and receiving the reward and the next state, and stores its experience to the shared common replay buffer $\mathcal{D}$. Then, the $i$-th learner updates its value function parameter $\theta^i$ by minimizing its own value loss function $\tilde{L}(\theta^i)$ which is the same as that of the base algorithm, and updates the policy parameter $\phi^i$ by minimizing the augmented loss function $\tilde{L}(\phi^i)$ in (1) for $N$ times by drawing $N$ mini-batches from the shared common replay buffer $\mathcal{D}$. Whenever an episode ends for a learner, the learner reports the episodic reward to the chief. The $i$-th learner reports $\hat{\mathbb{E}}_{s\sim\mathcal{D}}\left[D(\pi_{\phi^i}, \pi_{\phi^b})\right]$ to the chief for computation of $\hat{D}_{best}$ in (3). At every $M$ time steps, the chief updates $\beta$ according to (2), determines the best learner over the most recent $M$ time steps based on the collected episodic rewards from each learner. Once the best learner is determined, the chief obtains the policy parameter information $\phi^b$ from the determined best learner, and distributes the new $\beta$ and the reference policy parameter $\phi^b$ to all $N$ learners. This procedure repeats until the time steps reaches the predefined maximum. When the parallel learning based IPE reaches a steady state, we can choose any of the $N$ learners' policies and use the chosen policy for

the environment $\mathcal{E}$ in future since it is expected that at the steady-state the performance of all $N$ policies is more or less similar due to their distance property.

## 3.2 IPE-ENHANCED ALGORITHMS

The proposed IPE method can be applied to any off-policy RL algorithms regardless of whether the base RL algorithms have continuous actions or discrete actions. Here, we consider the application of IPE to the TD3 algorithm as the base algorithm and the constructed algorithm is named the IPE-TD3 algorithm. The details of baseline TD3 are explained in Appendix A. With TD3 as the base algorithm, each learner has its own parameters $\theta_1^i$, $\theta_2^i$, and $\phi^i$ for its two Q-functions and policy. Furthermore, it has $(\theta_1^i)'$, $(\theta_2^i)'$, and $(\phi^i)'$ which are the parameters of the corresponding target networks. For the distance measure between two policies, we use the mean square difference given by

$$D(\pi(s), \tilde{\pi}(s)) = \frac{1}{2} \|\pi(s) - \tilde{\pi}(s)\|_2^2. \tag{4}$$

For the $i$-th learner, as in TD3, the parameters $\theta_j^i$, $j = 1, 2$ are updated every time step by minimizing

$$\tilde{L}(\theta_j^i) = \hat{\mathbb{E}}_{(s,a,r,s') \sim \mathcal{D}} \left[ (y - Q_{\theta_j^i}(s, a))^2 \right] \tag{5}$$

where $y = r + \gamma \min_{j=1,2} Q_{(\theta_j^i)'}(s', \pi_{(\phi^i)'}(s') + \epsilon)$, $\epsilon \sim \text{clip}(\mathcal{N}(0, \tilde{\sigma}^2), -c, c)$. The parameter $\phi^i$ is updated every $d$ time steps by minimizing the following augmented loss function:

$$\tilde{L}(\phi^i) = \hat{\mathbb{E}}_{s \sim \mathcal{D}} \left[ -Q_{\theta_1^i}(s, \pi_{\phi^i}(s)) + \mathbf{1}_{\{i \neq b\}} \frac{\beta}{2} \left\| \pi_{\phi^i}(s) - \pi_{\phi^b}(s) \right\|_2^2 \right]. \tag{6}$$

For the first $T_{initial}$ timesteps for initial exploration we use a random policy and do not update all policies over the initial exploration period. With these loss functions, the reference policy, and the initial exploration policy, all procedure is the same as the general IPE procedure described previously. The pseudocode of the IPE-TD3 algorithm is provided in Appendix B.

The application of IPE to other algorithms such as SAC and DQN is also provided in Appendices.

## 4 EXPERIMENTS

In this section, we provide the numerical results on the performance of the proposed IPE-TD3 and current state-of-the-art on-policy and off-policy baseline algorithms on several MuJoCo environments (Todorov et al. (2012)). The baseline algorithms are Proximal Policy Optimization (PPO) (Schulman et al. (2017)), Actor Critic using Kronecker-Factored Trust Region (ACKTR) (Wu et al. (2017)), Soft Q-learning (SQL) (Haarnoja et al. (2017)), Soft Actor-Critic (SAC) (Haarnoja et al. (2018)), and TD3 (Fujimoto et al. (2018)). More numerical result on IPE applied to SAC is provided in Appendices.

### 4.1 PARAMETER SETTING

All hyperparameters we used for evaluation are the same as those in the original papers (Schulman et al. (2017); Haarnoja et al. (2018); Fujimoto et al. (2018)). Here, we provide the hyperparameters of TD3 and IPE-TD3 only.

**TD3** The networks for two Q-functions and the policy have 2 hidden layers. The first and second layers have sizes 400 and 300, respectively. The non-linearity function of the hidden layers is ReLU, and the activation functions of the last layers of the Q-functions and the policy are linear and hyperbolic tangent, respectively. We used the Adam optimizer with learning rate $10^{-3}$, discount factor $\gamma = 0.99$, target smoothing factor $\tau = 5 \times 10^{-3}$, the period $d = 2$ for updating the policy. The experience replay buffer size is $10^6$, and the mini-batch size $B$ is 100. The standard deviation for exploration noise $\sigma$ and target noise $\tilde{\sigma}$ are 0.1 and 0.2, respectively, and the noise clipping factor $c$ is 0.5.

**IPE-TD3** In addition to the parameters for TD3, we used $N = 4$ learners, the period $M = 250$ of updating the best policy and $\beta$, the number of recent episodes $E_r = 10$ for determining the best

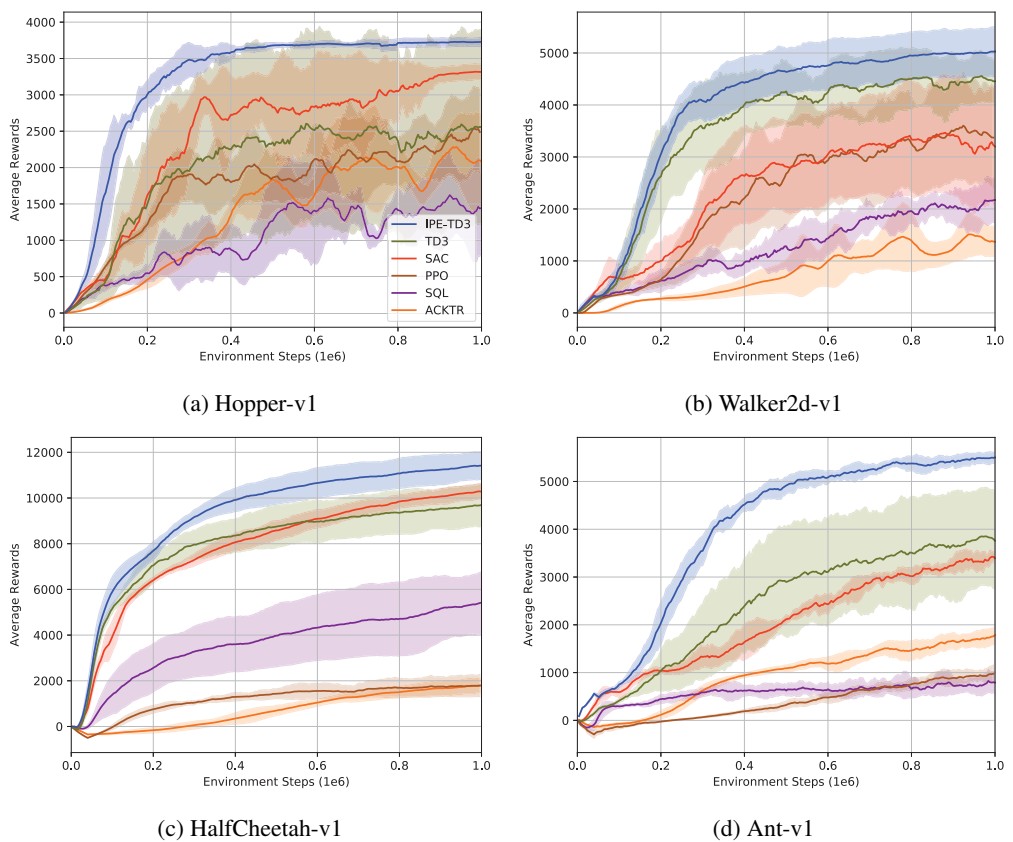

Figure 3: Performance for PPO (brown), ACKTR (orange), SQL (purple), SAC (red), TD3 (green), and IPE-TD3 (proposed method, blue) on MuJoCo tasks.

learner $b$. The parameters $d_{search}$ and $\rho$ for the exploration range are 0.04 and 2, respectively. The timesteps for initial exploration $T_{initial}$ is set as 250 for Hopper-v1 and Walker2d-v1 and as 2500 for HalfCheetah-v1 and Ant-v1.

## 4.2 COMPARISON TO BASELINES

In order to have sample-wise fair comparison among the considered algorithms, we obtain the performance with respect to environment steps (not time steps), which is defined as the total number of interactions with the environment by the agent. This comparison makes sense because the performance at the same environment steps means that all algorithms use the same number of samples obtained from the environment. The performance is obtained through the evaluation method that is similar to those in (Haarnoja et al. (2018); Fujimoto et al. (2018)). Evaluation of the policies are conducted every $R_{eval} = 4000$ environment steps for all algorithms. At each evaluation instant, the agent (or learner) fixes its policy as the one at the evaluation instant, and interacts with the same environment separate for the evaluation purpose with the fixed policy to obtain 10 episodic rewards. The average of these 10 episodic rewards is the performance at the evaluation instant. In the case of IPE-TD3 and other parallel learning schemes, each of the $N$ learners fixes its policy as the one at the evaluation instant, and interacts with the environment with the fixed policy to obtain 10 episodic rewards. First, the 10 episodic rewards are averaged for each learner and then the maximum of the 10-episode-average rewards of the $N$ learners is taken as the performance at that evaluation instant. We performed this operation for five different random seeds, and the mean and variance of the learning curve are obtained from these five simulations. The policies used for evaluation are stochastic for PPO and deterministic for the others.

Fig. 3 shows the learning curves over one million environment steps for several MuJoCo tasks: Hopper-v1, Walker2d-v1, HalfCheetah-v1, and Ant-v1. First, it is observed that the performance of TD3 here is similar to that in the original TD3 paper (Fujimoto et al. (2018)), and the performance of other baseline algorithms is also similar to that in the original papers (Schulman et al. (2017); Haarnoja et al. (2018)). It is seen that the IPE-TD3 algorithm outperforms the state-of-the-art RL algorithms in terms of both the speed of convergence with respect to environment steps and the final steady-state performance (except in Walker2d-v1, the initial convergence is a bit slower than TD3.) Especially, in the cases of Hopper-v1 and Ant-v1, TD3 has large variance and this means that the performance of TD3 is quite dependent on the initial condition of the environment and it is not easy for TD3 to escape out of bad local minima in certain environments. However, it is seen that IPE-TD3 yields much less variance as compared to TD3. This implies that the wide area search by IPE in the policy parameter space helps the learners escape out of bad local optima. It is seen that the wide area search around the previous best policy point in the policy parameter space by IPE yields faster and better policy search.

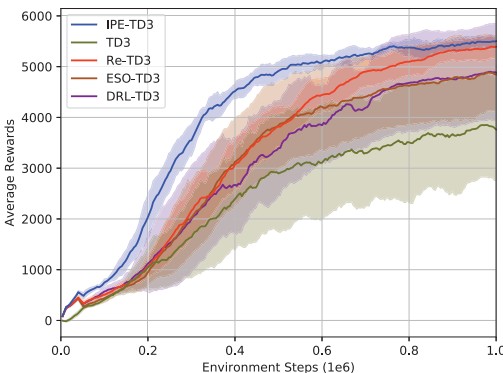

Figure 4: Ablation study: Different parallel methods for Ant-v1 task

## 4.3 ABLATION STUDY

IPE-TD3 has several components to improve the performance based on parallelism: 1) sharing experiences from multiple policies, 2) using the best policy information, and 3) fusing the best policy information in a soft manner based on the augmented loss function. Thus, we investigated the impact of each component on the performance improvement. For comparison we considered the following parallel policy exploration methods gradually incorporating more techniques:

1. **TD3** The original TD3 with one learner

2. **Distributed RL TD3 (DRL-TD3)** $N$ actors obtain samples from $N$ environment copies. The policy and the experience replay buffer are shared by all $N$ actors.

3. **Experience-Sharing-Only TD3 (ESO-TD3)** $N$ learners interact with $N$ environment copies and update their own policies using experiences drawn from the shared experience replay buffer.

4. **Reloading TD3 (Re-TD3)** At every $M'$ timesteps, the best policy is determined and all policies are initialized as the best policy, i.e., the best learner's policy parameter is copied to all other learners. The rest of the procedure is the same as experience-sharing-only TD3.

5. **IPE-TD3** At every $M$ timesteps, the best policy information is determined and this policy is used in a soft manner based on the augmented loss function.

Note that Re-TD3 exploits the best policy information from $N$ learners. The main difference between IPE-TD3 and Re-TD3 is the way how the best learner's policy parameter is used. Re-TD3 initializes all policies with the best policy parameter every $M'$ timesteps like in PBT (Jaderberg et al. (2017)), whereas IPE-TD3 uses the best learner's policy parameter information determined every $M$ timesteps to construct an augmented loss function. For fair comparison, $M$ and $M'$ are determined

Table 1: Steady state performance of different parallel exploration methods: IPE-TD3, Re-TD3, ESO-TD3, DRL-TD3 and TD3

| Environment | IPE-TD3 | Re-TD3 | ESO-TD3 | DRL-TD3 | TD3 |
|---|---|---|---|---|---|
| Hopper-v1 | 3729.92 | 3619.04 | **3745.77** | 3456.30 | 2555.85 |
| Walker2d-v1 | **5029.22** | 4921.51 | 4677.01 | 4813.72 | 4455.51 |
| HalfCheetah-v1 | 11418.50 | **11549.11** | 11086.88 | 11159.92 | 9695.92 |
| Ant-v1 | **5501.87** | 5393.09 | 4847.34 | 4885.74 | 3760.50 |

independently and optimally for IPE-TD3 and Re-TD3, respectively, since the optimal period can be different for the two methods. Thus, $M' = 5000$ is determined for Re-TD3 by tuning, whereas $M = 250$ is used for IPE-TD3. Since all $N$ policies collapse as one point in Re-TD3 at the beginning of each period, we expect that a larger period is required for Re-TD3 to have sufficiently spread policies at the end of the best policy selection period.

Fig. 4 shows the learning curves of the considered parallel exploration methods for the Ant-v1 task and Table 1 shows the final (steady-state) performance of the considered parallel exploration methods for four MuJoCo tasks, respectively. It is seen that IPE-TD3 outperforms other parallel methods: DRL-TD3, ESO-TD3 and Re-TD3 except the case that ESO-TD3 outperforms all other parallel schemes in Hopper-v1. Both Re-TD3 and IPE-TD3 have better final (steady-state) performance than TD3 and ESO-TD3 for all tasks except Hopper-v1 for which ESO-TD3 performs best. Note that ESO-TD3 obtains most diverse experiences since the $N$ learners shares the experience replay buffer but there is no interaction among the $N$ learners until the end of training. So, it seems that this diverse experience is beneficial to Hopper-v1. The final performances of Re-TD3 and IPE-TD3 are more or less the same for HalfCheetah-v1 but the final performance of IPE-TD3 is noticeably better than that of Re-TD3 in other cases.

## 5 CONCLUSION

In this paper, we have proposed a new interactive parallel learning scheme, IPE, to enhance the performance of off-policy RL systems. In the proposed IPE scheme, multiple identical learners with their own value-functions and policies sharing a common experience replay buffer search a good policy with the guidance of the best policy information in the previous search interval. The information of the best policy parameter of the previous search interval is fused in a soft manner by constructing an augmented loss function for policy update to enlarge the overall search space by the multiple learners. The guidance by the previous best policy and the enlarged search space by IPE enables faster and better policy search in the policy parameter space. The IPE-enhanced algorithms constructed by applying the proposed IPE scheme to TD3 or SAC outperforms most of the current state-of-the-art continuous-action RL algorithms. Although we mainly considered continuous-action off-policy algorithms in this paper, the proposed IPE method can also be applied to RL with discrete actions, as seen in Appendix E. In the case of continuous action control, the gain by IPE can be beneficial for recent trend of fast computer-based prototyping of complex robotics systems or autonomous cars, whereas in the discrete-action case better policy parameters can be searched for more challenging tasks by IPE.

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

## APPENDIX A. THE TWIN DELAYED DEEP DETERMINISTIC POLICY GRADIENT ALGORITHM AND THE SOFT ACTOR-CRITIC ALGORITHM

### A.1 THE TWIN DELAYED DETERMINISTIC POLICY GRADIENT (TD3) ALGORITHM

The TD3 algorithm is a current state-of-the-art off-policy algorithm and is a variant of the deep deterministic policy gradient (DDPG) algorithm (Lillicrap et al. (2015)). The TD3 algorithm tries to resolve two problems in typical actor-critic algorithms: 1) overestimation bias and 2) high variance in the approximation of the Q-function. In order to reduce the bias, the TD3 considers two Q-functions and uses the minimum of the two Q-function values to compute the target value, while in order to reduce the variance in the gradient, the policy is updated less frequently than the Q-functions. Specifically, let $Q_{\theta_1}, Q_{\theta_2}$ and $\pi_\phi$ be two current Q-functions and the current deterministic policy, respectively, and let $Q_{\theta'_1}, Q_{\theta'_2}$ and $\pi_{\phi'}$ be the target networks of $Q_{\theta_1}, Q_{\theta_2}$ and $\pi_\phi$, respectively. The target networks are initialized by the same networks as the current networks. At time step $t$, the TD3 algorithm takes an action $a_t$ with exploration noise $\epsilon$: $a_t = \pi_\phi(s_t) + \epsilon$, where $\epsilon$ is zero-mean Gaussian noise with variance $\sigma^2$, i.e., $\epsilon \sim \mathcal{N}(0, \sigma^2)$. Then, the environment returns reward $r_t$ and the state is switched to $s_{t+1}$. The TD3 algorithm stores the experience $(s_t, a_t, r_t, s_{t+1})$ at the experience replay buffer $\mathcal{D}$. After storing the experience, the Q-function parameters $\theta_1$ and $\theta_2$ are updated by gradient descent of the following loss functions:

$$L(\theta_j) = \hat{\mathbb{E}}_{(s,a,r,s') \sim \mathcal{D}} \left[ (y - Q_{\theta_j}(s,a))^2 \right], \quad j = 1, 2 \tag{7}$$

where $\hat{\mathbb{E}}_{(s,a,r,s') \sim \mathcal{D}}$ denotes the sample expectation with an uniform random mini-batch of size $B$ drawn from the replay buffer $\mathcal{D}$, and the target value $y$ is given by

$$y = r + \gamma \min_{j=1,2} Q_{\theta'_j}(s', \pi_{\phi'}(s') + \epsilon), \quad \epsilon \sim \text{clip}(\mathcal{N}(0, \tilde{\sigma}^2), -c, c). \tag{8}$$

Here, for the computation of the target value, the minimum of the two target Q-functions is used to reduce the bias. The procedure of action taking and gradient descent for $\theta_1$ and $\theta_2$ are repeated for $d$ times ($d = 2$), and then the policy and target networks are updated. The policy parameter $\phi$ is updated by gradient descent by minimizing the loss function for $\phi$:

$$L(\phi) = -\hat{\mathbb{E}}_{s \sim \mathcal{D}} \left[ Q_{\theta_1}(s, \pi_\phi(s)) \right], \tag{9}$$

and the target network parameters $\theta'_j$ and $\phi'$ are updated as

$$\theta'_j \leftarrow (1 - \tau)\theta'_j + \tau\theta_j \qquad \phi' \leftarrow (1 - \tau)\phi' + \tau\phi. \tag{10}$$

The networks are trained until the number of time steps reaches a predefined maximum.

### A.2 THE SOFT ACTOR-CRITIC (SAC) ALGORITHM

The SAC algorithm is an off-policy algorithm comparable to TD3 and yields good performance especially in environments with high dimensional action spaces. The SAC algorithm is a maximum entropy RL which is based on the discounted sum of reward and the entropy of the current policy given by

$$\mathbb{E}_{\tau \sim \pi} \left[ \sum_{t=0}^{\infty} \gamma^t \left( r(s_t, a_t) + \alpha \mathcal{H}(\pi(\cdot|s_t))) \right) \right], \tag{11}$$

where $\alpha$ is a weighting factor that balances between the reward and the entropy of the policy. This objective function stimulates the algorithm to explore more diverse experiences so as to find a better policy.

The SAC algorithm has one value function $V_\psi(s)$, two Q-functions $Q_{\theta_j}(s, a), j = 1, 2$, and one stochastic policy $\pi_\phi(\cdot|s)$, which are parameterized by parameters $\psi, \theta_j$, and $\phi$, respectively. It also has a target value function $V_{\psi'}(s)$ for stable convergence.

After initialization, at each time step $t$ the algorithm obtains experience $(s_t, a_t, r_t, s_{t+1})$ by interacting with the environment and stores the experience to the experience replay buffer $\mathcal{D}$. Then, it

updates the parameters $\psi$, $\theta_j$, and $\phi$ by gradient descent of the following loss functions:

$$J(\psi) = \hat{\mathbb{E}}_{s\sim\mathcal{D},a\sim\pi_\phi(\cdot|s)} \left[\frac{1}{2}\left\|V_\psi(s) - \bar{Q}(s,a) + \log\pi_\phi(a|s)\right\|_2^2\right], \tag{12}$$

$$J(\theta_j) = \hat{\mathbb{E}}_{(s,a,r,s')\sim\mathcal{D}} \left[\frac{1}{2}\left(Q_{\theta_j}(s,a) - r/\alpha - \gamma V_{\psi'}(s')\right)^2\right], \quad j = 1, 2 \tag{13}$$

$$J(\phi) = \hat{\mathbb{E}}_{s\sim\mathcal{D},a\sim\pi_\phi(\cdot|s)} \left[\log\pi_\phi(a|s) - \bar{Q}(s,a)\right], \tag{14}$$

where $\bar{Q}(s,a) = \min\{Q_{\theta_1}(s,a), Q_{\theta_2}(s,a)\}$, and $\hat{\mathbb{E}}_{(s,a,r,s')\sim\mathcal{D}}$ is the sample expectation with an uniform random mini-batch of size $B$ drawn from the replay buffer $\mathcal{D}$. After updating the parameters, the target value function parameter $\psi'$ is updated as

$$\psi' \leftarrow (1 - \tau)\psi' + \tau\psi \tag{15}$$

In order to obtain diverse experience in the initial stage of learning, it uses a uniform policy for initial $T_{initial}$ time steps and the current policy $\pi_\phi(\cdot|s)$ for the rest of learning.

## APPENDIX B. PSEUDOCODE OF THE IPE-TD3 ALGORITHM

---

**Algorithm 1** The Interactive Parallel Exploration TD3 (IPE-TD3) Algorithm

---

**Require:** $N$: number of learners, $T_{initial}$: initial exploration time steps, $T$: maximum time steps, $M$ : the best-policy update period, $B$: size of mini-batch, $d$: update interval for policy and target networks.

1: Initialize $\phi^1 = \cdots = \phi^N = \phi^b$, $\theta_j^1 = \cdots = \theta_j^N$, $j = 1, 2$, randomly.
2: Initialize $\beta = 1, t = 0$
3: **while** $t < T$ **do**
4:     $t \leftarrow t + 1$ (one time step)
5:     **for** $i = 1, 2, \cdots, N$ in parallel **do**
6:         **if** $t < T_{initial}$ **then**
7:             Take a uniform random action $a_t^i$ to environment copy $\mathcal{E}^i$
8:         **else**
9:             Take an action $a_t^i = \pi^i\left(s_t^i\right) + \epsilon$, $\epsilon \sim \mathcal{N}(0, \sigma^2)$ to environment copy $\mathcal{E}^i$
10:         **end if**
11:         Store experience $(s_t^i, a_t^i, r_t^i, s_{t+1}^i)$ to the shared common experience replay $\mathcal{D}$
12:     **end for**
13:     **if** $t < T_{initial}$ **then**
14:         **continue** (i.e., go to the beginning of the while loop)
15:     **end if**
16:     **for** $i = 1, 2, \cdots, N$ in parallel **do**
17:         **for** $k = 1, 2, \cdots, N$ **do**
18:             Sample a mini-batch $\mathcal{B} = \{(s_{t_l}, a_{t_l}, r_{t_l}, s_{t_l+1})\}_{l=1,\ldots,B}$ from $\mathcal{D}$
19:             Update $\theta_j^i$, $j = 1, 2$, by gradient descent for minimizing $\tilde{L}(\theta_j^i)$ in (5) with $\mathcal{B}$
20:             **if** $k \equiv 0(\mathrm{mod}\ d)$ **then**
21:                 Update $\phi^i$ by gradient descent for minimizing $\tilde{L}(\phi^i)$ in (6) with $\mathcal{B}$
22:                 Update the target networks: $(\theta_j^i)' \leftarrow (1 - \tau)(\theta_j^i)' + \tau\theta_j^i$, $(\phi^i)' \leftarrow (1 - \tau)(\phi^i)' + \tau\phi^i$
23:             **end if**
24:         **end for**
25:     **end for**
26:     **if** $t \equiv 0(\mathrm{mod}\ M)$ **then**
27:         Select the best learner $b$
28:         Adapt $\beta$ with (2)
29:     **end if**
30: **end while**

---

APPENDIX C. PSEUDOCODE OF THE IPE-SAC ALGORITHM

---

**Algorithm 2** The Interactive Parallel Exploration SAC (IPE-SAC) Algorithm

---

**Require:** $N$: number of learners, $T_{initial}$: initial exploration time steps, $T$: maximum time steps, $M$ : the best-policy update period, $B$: size of mini-batch
1: Initialize $\psi^1 = \cdots = \psi^N$, $\phi^1 = \cdots = \phi^N = \phi^b$, $\theta_j^1 = \cdots = \theta_j^N$, $j = 1, 2$, randomly.
2: Initialize $\beta = 1$, $t = 0$
3: **while** $t < T$ **do**
4:   $t \leftarrow t + 1$ (one time step)
5:   **for** $i = 1, 2, \cdots, N$ in parallel **do**
6:     **if** $t < T_{initial}$ **then**
7:       Take a uniform random action $a_t^i$ to environment copy $\mathcal{E}^i$
8:     **else**
9:       Take an action $a_t^i \sim \pi^i \left( \cdot | s_t^i \right)$ to environment copy $\mathcal{E}^i$
10:     **end if**
11:     Store experience $(s_t^i, a_t^i, r_t^i, s_{t+1}^i)$ to the shared common experience replay $\mathcal{D}$
12:   **end for**
13:   **if** $t < T_{initial}$ **then**
14:     **continue** (i.e., go to the beginning of the while loop)
15:   **end if**
16:   **for** $i = 1, 2, \cdots, N$ in parallel **do**
17:     **for** $k = 1, 2, \cdots, N$ **do**
18:       Sample a mini-batch $\mathcal{B} = \{(s_{t_l}, a_{t_l}, r_{t_l}, s_{t_l+1})\}_{l=1,\dots,B}$ from $\mathcal{D}$
19:       Update $\psi^i$, $\theta_j^i$, and $\phi^i$ by gradient descent for minimizing (16), (17), and (18) with $\mathcal{B}$, respectively.
20:       Update the target parameters: $(\psi^i)' \leftarrow (1 - \tau)(\psi^i)' + \tau \psi^i$
21:     **end for**
22:   **end for**
23:   **if** $t \equiv 0 \pmod{M}$ **then**
24:     Select the best learner $b$
25:     Update the best policy parameter $\phi^b$
26:     Adapt $\beta$ with (2)
27:   **end if**
28: **end while**

---

In IPE-SAC, each learner has its own parameters $\psi^i$, $\theta_1^i$, $\theta_2^i$, and $\phi^i$ for its value function, two Q-functions, and policy. Each learner also has $(\psi^i)'$ which is the parameter of the target value function. For the distance measure between two policies, we use the mean square difference of the mean action of Gaussian policy, given by $D(\pi(\cdot|s), \tilde{\pi}(\cdot|s)) = \frac{1}{2} \|\text{mean}\{\pi(\cdot|s)\} - \text{mean}\{\tilde{\pi}(\cdot|s)\}\|_2^2$. The $i$-th learner updates the parameters $\psi^i$, $\theta_1^i$, $\theta_2^i$, and $\phi^i$ every timestep by minimizing

$$\tilde{L}(\psi^i) = \hat{\mathbb{E}}_{s\sim\mathcal{D}, a\sim\pi_{\phi^i}(\cdot|s)} \left[ \frac{1}{2} \left\| V_{\psi^i}(s) - \bar{Q}^i(s, a) + \log \pi_{\phi^i}(a|s) \right\|_2^2 \right] \tag{16}$$

$$\tilde{L}(\theta_j^i) = \hat{\mathbb{E}}_{(s,a,r,s')\sim\mathcal{D}} \left[ \frac{1}{2} \left( Q_{\theta_j^i}(s, a) - r/\alpha - \gamma V_{(\psi^i)'}(s') \right)^2 \right], \quad j = 1, 2 \tag{17}$$

$$\tilde{L}(\phi^i) = \hat{\mathbb{E}}_{s\sim\mathcal{D}, a\sim\pi_{\phi^i}(\cdot|s)} \Big[ \log \pi_{\phi^i}(a|s) - \bar{Q}^i(s, a)$$
$$+ \mathbf{1}_{\{i \neq b\}} \frac{\beta}{2} \left\| \text{mean}\{\pi_{\phi^i}(\cdot|s)\} - \text{mean}\{\pi_{\phi^b}(\cdot|s)\} \right\|_2^2 \Big] \tag{18}$$

where $\bar{Q}^i(s, a) = \min\left\{ Q_{\theta_1^i}(s, a), Q_{\theta_2^i}(s, a) \right\}$. After updating these parameters, each learner updates its target value function parameters. With these loss functions, all procedure is the same as the general IPE procedure described in Section 3. The pseudocode of the IPE-SAC algorithm is shown above.

## APPENDIX D. RESULT OF IPE-SAC ON HUMANOID (RLLAB)

As mentioned already, IPE is general in that it can be applied to other off-policy algorithms. Here, we provide numerical results on IPE-SAC, shown in Appendix C, constructed by combining IPE with SAC. Experiment was perform on the task of Humanoid (rllab) (Duan et al. (2016)) that needs more exploration. We compared IPE-SAC with SAC and multi-learner reloading SAC (Re-SAC) which copies the parameter of the best learner to other learners periodically.

### D.1 PARAMETER SETTING

**SAC** The networks for the state-value function, two Q-functions, and the policy had 2 hidden layers of size 256. The activation functions for the hidden layers and the last layers were ReLU and linear, respectively. We used the Adam optimizer with learning rate $3 \times 10^{-4}$, discount factor $\gamma = 0.99$, and target smoothing factor $\tau = 5 \times 10^{-3}$. The algorithm was trained by random mini-batches of size $B = 256$ from the experience replay buffer of the maximum size $10^6$. The reward scale for updating Q-functions was 10 for the Humanoid (rllab) environment. The initial exploration timesteps $T_{initial}$ was set to 1000.

**IPE-SAC** Additional parameters for IPE-SAC are as follows. We used $N = 4$ learners, the period $M = 500$ of updating the best policy and $\beta$, the number of recent episodes $E_r = 10$ for determining the best learner $b$. The parameters $d_{search}$ and $\rho$ for the exploration range were 0.01 and 2, respectively. We used the initial exploration timesteps $T_{initial} = 250$.

### D.2 PERFORMANCE ON HUMANOID (RLLAB)

The learning curve on Humanoid (rllab) is shown in Figure 5[1]. It is seen that IPE-SAC outperforms the original SAC and Re-SAC. This result shows the promising aspect of IPE that it can be useful for tasks requiring more exploration.

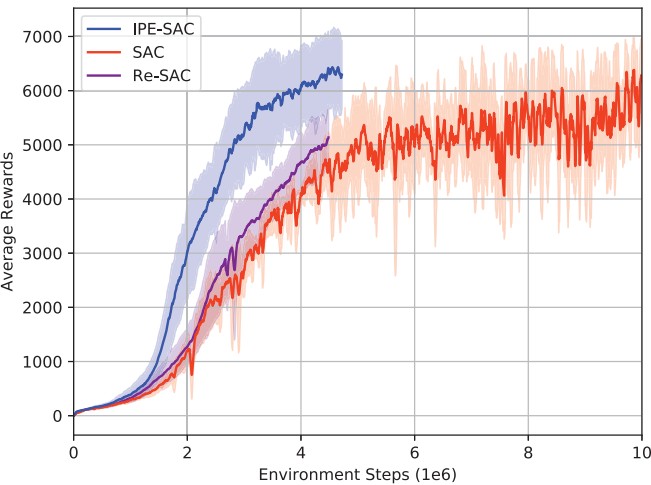

Figure 5: Performance of IPE-SAC (blue), Re-SAC (purple), and SAC (red) on Humanoid (rllab)

---

[1]The simulation is still running, and we will change the graph when the simulation is finished.

APPENDIX E. PSEUDOCODE OF THE IPE-DQN ALGORITHM

---

**Algorithm 3** The Interactive Parallel Exploration DQN (IPE-DQN) Algorithm

---

**Require:** $N$: number of learners, $T_{initial}$: initial exploration time steps, $T$: maximum time steps, $M$: the best-policy update period, $B$: size of mini-batch, $f$: update interval for Q-functions, $d$: update interval for target Q-functions,.
1: Initialize $\theta^1 = \cdots = \theta^N = \theta^b$ randomly.
2: Initialize $\beta = 1, t = 0$
3: **while** $t < T$ **do**
4:     $t \leftarrow t + 1$ (one time step)
5:     **for** $i = 1, 2, \cdots, N$ in parallel **do**
6:       **if** $t < T_{initial}$ **then**
7:         Take a uniform random action $a_t^i$ to environment copy $\mathcal{E}^i$
8:       **else**
9:         Take an action $a_t^i = \arg\max_{a \in \mathcal{A}} \left\{ Q_{\theta^i}(s_t^i, a) \right\}$ w.p. $1 - \varepsilon$ or a uniform random action $a_t^i$ w.p. $\varepsilon$ to environment copy $\mathcal{E}^i$
10:       **end if**
11:       Store experience $(s_t^i, a_t^i, r_t^i, s_{t+1}^i)$ to the shared common experience replay $\mathcal{D}$
12:     **end for**
13:     **if** $t < T_{initial}$ **then**
14:       **continue** (i.e., go to the beginning of the while loop)
15:     **end if**
16:     **for** $i = 1, 2, \cdots, N$ in parallel **do**
17:       **for** $k = 1, 2, \cdots, N$ **do**
18:         **if** $k \equiv 0 (\bmod f)$ **then**
19:          Sample a mini-batch $\mathcal{B} = \{(s_{t_l}, a_{t_l}, r_{t_l}, s_{t_l+1})\}_{l=1,\ldots,B}$ from $\mathcal{D}$
20:          Update $\theta^i$ by gradient descent minimizing (19) with $\mathcal{B}$.
21:         **end if**
22:       **end for**
23:     **end for**
24:     **if** $t \equiv 0 (\bmod d)$ **then**
25:       **for** $i = 1, 2, \cdots, N$ in parallel **do**
26:         Update $(\theta^i)' \leftarrow \theta^i$
27:       **end for**
28:     **end if**
29:     **if** $t \equiv 0 (\bmod M)$ **then**
30:       Select the best learner $b$
31:       Update the best policy parameter $\theta^b$
32:       Adapt $\beta$ with (2)
33:     **end if**
34: **end while**

---

IPE can also be applied to off-policy algorithms with discrete action spaces as well as continuous action spaces. Thus, we applied IPE to DQN to construct IPE-DQN. In IPE-DQN, each learner has its own Q-function parameters $\theta^i$ and target Q-function parameters $(\theta^i)'$. We define the distance for two Q-functions $Q(s, a)$ and $\tilde{Q}(s, a)$ as $D(Q(s, \cdot), \tilde{Q}(s, \cdot)) = \mathrm{KL}\left( \mathrm{softmax}\left( Q(s, \cdot) \right) || \mathrm{softmax}\left( \tilde{Q}(s, \cdot) \right) \right)$. We used the Q-function parameter $\theta^b$ of the best learner as the reference parameter, which was originally $\phi^b$ in (1) and (3). The $i$-th learner updates the parameters $\theta^i$ every $f$ timesteps by minimizing $\tilde{L}(\theta^i) =$

$$\hat{\mathbb{E}}_{(s,a,r,s') \sim \mathcal{D}} \left[ \frac{1}{2} \left\| Q_{\theta^i}(s, a) - y \right\|_2^2 + \mathbf{1}_{\{i \neq b\}} \beta \mathrm{KL}\left( \mathrm{softmax}\left( Q_{\theta^i}(s, \cdot) \right) || \mathrm{softmax}\left( Q_{\tilde{\theta}^b}(s, \cdot) \right) \right) \right] \tag{19}$$

where $y = r + \gamma Q_{(\theta^i)'}(s', \arg\max_{a' \in \mathcal{A}} \{Q_{\theta^i}(s', a')\})$. With the loss function and the reference policy, all procedure is the same as the general IPE procedure described in Section 3. The pseudocode of the IPE-DQN algorithm is shown above.

