# OpenReview forum: "Interactive Parallel Exploration for Reinforcement Learning in Continuous Action Spaces"
_ICLR.cc/2019/Conference_

### Official Review · AnonReviewer3 · 2018-10-28
**Decent results but incomplete comparisons**

**Rating:** 7
**Confidence:** 4

**Review:**

The paper present interactive parallel exploration (IPE), a reinforcement learning method based on an ensemble of policies and a shared experience pool. Periodically, the highest-return achieving policy is selected, towards which the other policies are updated in a sense of some distance metric. IPE is applicable to any off-policy reinforcement learning algorithm. The experiments demonstrate some improvement over TD3 on four MuJoCo benchmark tasks.

The method is motivated heuristically, and and it provides some benefits in terms of sample efficiency and lower variance between training trials. However, it is hard to justify the increased algorithmic complexity and additional hyperparameters just based on the presented results. The paper motivates IPE as an add-on that can increase the performance of any off-policy RL algorithm. As such, I would like to see IPE being applied to other algorithms (e.g., SAC or DQN) as a proof of generalizability, and compared to other similar ensemble based algorithms (e.g., bootstrapped DQN).

While the improvement in the sample complexity is quite marginal, what I find the most interesting is how IPE-TD3 reduces variance between training trials compared to vanilla TD3. Convergence to bad local optimum can be a big problem, and IPE could help mitigate it. I would suggest including environments where local optima can be a big problem, for example HumanoidStandup, or any sparse reward task. Also the paper does not include ablations, which, given the heuristic nature of the proposed method, seems important.

---

> ### Author Response · Authors · 2018-11-26
> **Response to Reviewer 3**
>
> Response to Application of IPE to Other Algorithms: During the revision, we applied IPE to SAC and DQN. The constructed IPE-SAC and IPE-DQN algorithms are provided in Appendices C and E. We applied IPE-SAC to a more challenging problem Humanoid and obtained some preliminary result (not full result due to time limitation). The preliminary experimental result shows the effectiveness of IPE on SAC. Please see Appendix D.
>
> We also applied IPE to DQN and constructed IPE-DQN, which is provided in Appendix E. We tried IPE-DQN on several Atari games but could not obtain numerical result within this short revision window. We used the DQN code provided by OpenAI baseline to construct IPE-DQN with multiple learners. Somehow the execution of this code is slow as compared to other IPE-enhanced algorithms. We need more debugging time for IPE-DQN.
>
> We also applied IPE-TD3 to a multi-agent RL environment since the IPE idea can directly be applied to multi-agent RL environments where information exchange during training is possible.
>
> Response to More Challenging Humanoid: During the revision, we considered Humanoid. For this, we constructed IPE-SAC (Appendix C) since TD3 does not learn properly Humanoid as seen in the original SAC paper. We applied IPE-SAC and obtained some preliminary result (not full result due to time limitation) for Humanoid. The result is shown in Appendix D of the revised paper. The preliminary result shows that the proposed IPE method seems effective in the task of Humanoid.
>
> Response to Previous Works: We have newly included many relevant references in Section 2 of the revised paper and explained the difference of the proposed method from the existing methods.

---

> > ### Comment · AnonReviewer3 · 2018-12-01
> > **Response**
> >
> > Thank you for the revised version. Especially the additional comparison on rllab Humanoid is really convincing, as it demonstrates that the method works with multiple off-policy algorithms, and the performance is substantially better than anything I have seen before. I believe accepting this paper for publication would be of great benefit to the deep RL community, and I have thus increased my score.

---

### Official Review · AnonReviewer1 · 2018-11-01
**A new parallel training architecture**

**Rating:** 4
**Confidence:** 4

**Review:**

This paper describes a new architecture for parallelizing off-policy reinforcement learning systems with a pool of independent learners trained on identical, but independent instances of the environment with a scheme for periodically synchronizing the the policy knowledge across the pool. The paper provides demonstrations in several continuous control domains.

I think this paper should be rejected because: 1) the approach is not well justified or placed within the large literature on parallel training architectures and population-based training methods, (2) the results are competitive with the best in each domain, but there are many missing details. Since the contribution is entirely support by empirical evidence, these issues need to be clarified. I look forward to the author response, as I will pose several questions below and my final score will carefully take the answers into account.

Justification of decision. There are numerous papers on parallel architectures for training deep RL systems [1,2, 6] and you cited a few, and while not all of them focus on continuous control there are design decisions and insights in those works must be relevant to your efforts. You should make those connections clear in the paper. One line of the paper is not nearly enough. The stated focus of the paper is exploration-exploitation yet there is little to no discussion of other ideas including noisy networks, intrinsic motivation, or count-based exploration methods. The paper is missing a lot of key connections to the literature.

I am certainly not a fan of architectures that assume access to many instances of the environment. In this case that assumption seems worse because of the target application: continuous control domains. These domains are simulations of physical control systems; on a robot the agent receives only one stream of experience and thus these architectures would not work well. Though there is some work on applying these multi-environment architectures to farms of robot arms; the reality of the real-world is that the arms end up being very different due to wear and tear, and engineers must constantly fix the hardware because these multi-environment architectures do not work when the environments are different. We cannot loose sight of the goal here—maximizing these simulation environments is not of interest itself, its a stepping stone—architectures that only work on simulations that afford multiple identical environments but fail in the real world have very limited application. I think this paper needs to motivate why parallel training in this way in these robotics inspired domains is interesting and worthwhile.

The main area of concern with this paper is the experiment section. There are several issues/questions I would like the authors to address:
1) You built on top of TD3, however you just used the parameter settings of TD3 as published and didn’t tune them. This is a problem because it could just be that the existing parameter choices for TD3 were just better for the new approach. You have to take additional effort in this case to ensure your method is actually better than just using TD3. Additional parameter tuning of TD3 is required here.
2) I think its an odd choice for TD3 to have an infinite buffer, as recent work has show at least for DQN that large buffers can hurt performance [7].  Can you justify this choice beyond “the authors of TD3 did it that way”?
3) Why is R_eval different for each method?
4) Why did you not compare to TD3 on the same set of domains as used in the TD3 paper? Why a subset? Why these particular domains?
5) In 2 of the 4 domains the proposed method ties or is worse than the baselines. In half-cheetah it looks close to significant, and in the ant domain the result is unlikely to be significant because the error-bars overlap and the error-bars of TD3 are wider than the other methods so a simple visual inspection is not enough. There does not seem to be a strong case for the new method here. I may be misunderstanding the results. Help me see the significance.
6) The paper claims improvement in variance, but this requires additional analysis in the form of an F-test of better.
7) Why these baselines (e.g., SAC) and not others? Why did you not include D4PG [6]? Soft Q-learning? A population-based training method [3,4,5,8] to name a few?

[1] GPU-Accelerated Robotic Simulation for Distributed Reinforcement Learning
[2] IMPALA: Scalable Distributed Deep-RL with Importance Weighted Actor-Learner Architectures
[3] Human-level performance in first-person multiplayer games with population-based deep reinforcement learning
[4] Improving Exploration in Evolution Strategies for Deep Reinforcement Learning via a Population of Novelty-Seeking Agents
[5] Structured Evolution with Compact Architectures for Scalable Policy Optimization
[6] Distributed Distributional Deterministic Policy Gradients
[7] A Deeper Look at Experience Replay
[8] Evolution Strategies as a Scalable Alternative to Reinforcement Learning


Small things that did not impact the score:
1) references to “search interval” in the abstract are confusing because the reader has not read the paper yet
2) Description of the method in abstract is too specific
3) P1 intro, not a topic sentence for what follows
4) “performs an action to its environment” >> grammar
5) “One way to parallel learning…” >> grammar
6) “that the value parameter and” >> grammar
7) “pitfalls” >> minima
8) Did you try combining you method with other base off-policy methods? how did it work?
9) GAE undefined?
10) “among the baseline and”>>grammar…there are many grammar errors

---

> ### Author Response · Authors · 2018-11-26
> **Response to Reviewer 1**
>
> Response to Relation with Previous Works: We agree with the reviewer in that the proposed method is not properly placed in the literature in the original paper. However, the proposed method has new ingredients in parallel learning, as mentioned by Reviewer 2. In Section 4.3 of the revised paper, one can clearly see the impact of the new ingredients in the context of the previous works. During the revision, we did extensive literature survey and framed our work in the context of the previous works including the references that the reviewer mentioned. Please see Section 2 of the revised paper.
>
> Response to Missing Details: We agree with the reviewer in that there were many things missing in the original paper. We included relevant new results in the revised paper. Please see Section 2, Section 4.3 and Appendices. Please see the response to all reviewers for what we have done during the revision.
>
> Response to Motivation in Parallel Learning in Continuous-Action RL: Please see the response to all reviewers. Indeed, fast prototyping for development of large complex robot systems based on precise Newtonian-dynamics simulation platforms becomes more and more important to alleviate the difficulties that the reviewer mentioned. Please visit https://cyberbotics.com/ .  Furthermore, the proposed IPE method can be applied to off-policy RL algorithms with discrete actions such as DQN. The constructed IPE-DQN algorithm is shown in Appendix E of the revised paper. We tried IPE-DQN on several Atari games but could not obtain numerical result in this short revision time. We have added a few sentences regarding this motivation in the conclusion of the revised paper.
>
> Response to Parameter Setting of TD3: Based on common sense, we believe that the authors of TD3 fine-tuned the parameters of their TD3. As shown in Appendix D of the revised paper, IPE-SAC also performs better than SAC with the parameters in the SAC paper. This is another evidence of generality of IPE.
>
> Response to The Infinite Buffer Size of TD3: The simulation is over 1 million environment steps, so the buffer size is actually 1 million. For clarity in the revised paper we explicitly set the size of replay buffer as 1 million and obtained new performance plots in the revised paper.
>
> Response to R_eval: R_eval is the frequency of measuring the performance, and it does not affect training at all. The reason for different R_eval is that the implementations of algorithms have different frequencies for writing logs. This can be modified easily, so we set the R_eval = 4000 environment steps for all algorithms in the revised paper.
>
> Response to Other Environments for TD3: The environments (Reacher-v1, InvertedPendulum-v1, InvertedDoublePendulum-v1) were not considered because the performance of the previous algorithms on these environments already saturated, as seen in the original TD3 paper (Fujimoto et al., 2018). It seems that they achieved the optimal performance already. IPE-TD3 will not provide improvement on these environments over other algorithms. So, we did not feel the necessity for test of IPE-TD3 on these tests. What we need is more challenging tasks.
>
> Response to Significance of IPE: The two of the four domains in the paper are easy tasks so that other algorithms already work well. What we need is more challenging tasks to see the real gain of the method, as mentioned by Reviewers 2 and 3. In more challenging Humanoid suggested by other reviewers, the constructed IPE-SAC outperforms SAC, as seen in Appendix D. Here, TD3 does not ever work.
>
> Response to F-test: F-test for variance is conducted with null hypothesis var_TD3 = var_IPE-TD3 and alternative hypothesis var_TD3 > var_IPE-TD3. The F-statistics are 280.37, 0.69, 2.73, and 9.45 for Hopper-v1, Walker2d-v1, HalfCheetah-v1, and Ant-v1, respectively. Thus, it seems that the variance of IPE-TD3 is smaller than that of TD3 in Hopper-v1 and Ant-v1 tasks.
>
> Response to Other Baselines: The revised paper includes more baselines (ACKTR and SQL). As shown Figure 3, IPE-TD3 outperforms all baselines in the four MuJoCo tasks. In ablation study of the revised paper, we considered other parallel enhancement methods. Among them, the reloading method is based on the idea in PBT (Jaderberg et al., 2017) and simply copies the best policy parameter to other learners. As seen in Table 1 and the figures, IPE outperforms the reloading method. So, it seems that the proposed way of fusing the best policy parameter is more effective than simply copying. Note that simple copying means that the search area covered by all learners collapses to one point at the time of copying. We don’t need to do this.
>
> Response to Minor Comments: Thanks for careful reading. We checked the grammar. Moreover IPE-enhanced algorithms do not use GAE.

---

### Official Review · AnonReviewer2 · 2018-11-02
**Potential for impact but not well framed in prior work and limited evaluation**

**Rating:** 6
**Confidence:** 4

**Review:**

Revision: The authors added many references to prior work to the paper and did some additional experiments that certainly improved the quality. However, the additional results also show that the shared experience buffer doesn't have that much influence and that for the original tasks (the humanoid results in the appendix look more promising but inconclusive) the reloading variant seems to catch up relatively quickly. Reloading and distributed learning seem to lead to the largest gains but those methods already existed. That said, the IPE method does give a clear early boost. It's not clear yet whether the method can also lead to better end results. I improved my score because I think that the idea and the results are worth sharing but I'm still not very convinced of their true impact yet.

The paper proposes a scheme for training multiple RL agents in parallel using a shared replay buffer and an objective that pulls the policies towards the best performing policy as determined by the last comparison event. The method is combined with the TD3 continuous control learning algorithm and evaluated on Mujoco tasks from OpenAI Gym.

The experiments in the paper seem correctly executed and it is nice that there are multiple baselines but I'm not convinced that the comparison is very insightful. It is somewhat odd that the architectures for the different methods differ quite a bit sometimes. The experiments are already hard to compare due to the very different natures of the optimization algorithms (distributed or not, asynchronous or not). It would be nice to also see plots of the results as a function of the number of learner steps and wall time if these can be obtained from the already executed experiments.

The paper doesn’t include many references and fails to mention research about parallel (hyperparameter) optimization methods that seems very related, even if the goal of those methods is not always behavioral exploration. Especially Population Based Training (PBT; Jaderberg et al., 2017) is very similar in spirit in that a population of parallel learners occasionally exchange information to benefit from the findings of the best performing individuals. The method is also similar to knowledge distillation (Hinton et al. 2015), which has also been used to speed up multi-task RL (Teh et al., 2017). It would also be nice to see an ablation of some of the different components of the algorithm. For example, it would be interesting to know how important the following of the best policy is in comparison to the gains that are obtained from simply using a shared replay buffer.

The paper is easy to follow and seems to describe the methods in enough detail to allow for a replication of the experiments. The terminology is not always precise and I’m a bit confused about whether the distance between policies is measured between their actions or their parameter vectors. Equation 8 suggests the former (as I'm assuming is what is also meant in the paper) but the text often speaks about the search radius in parameter space.

Exploration is a big problem in reinforcement learning and while parallelization of environment simulations helps to speed up training, additional computational effort typically provides diminishing returns. Methods for coordinating parallel exploration could have a severe impact. Since many RL setups are already distributed, the novelty of the paper mainly comes from sharing a replay buffer (I haven't seen this before but it seems like such an obvious thing to try that I wouldn't be surprised if it has been done) and the way in which learners are forced to follow the best individual. It is promising that the method provides the largest gains for the environment which seems to be the most challenging but it’s hard to draw conclusions from these results. It would be more insightful to see how the method performs on more challenging tasks where exploration is more important, but I understand that these experiments are computationally demanding.

All in all, the paper presents a method that is simple while having potential for impact but needs to frame it more in the context of previous work. The empirical evaluation is a bit limited and would be more impressive with some additional tasks or at least benefit from a more thorough analysis of the settings and relative contributions of the shared replay buffer and following of the best policy.

References
Jaderberg, M., Dalibard, V., Osindero, S., Czarnecki, W. M., Donahue, J., Razavi, A., ... & Fernando, C. (2017). Population based training of neural networks. arXiv preprint arXiv:1711.09846.
Hinton, G., Vinyals, O., & Dean, J. (2015). Distilling the knowledge in a neural network. arXiv preprint arXiv:1503.02531.
Teh, Y., Bapst, V., Czarnecki, W. M., Quan, J., Kirkpatrick, J., Hadsell, R., ... & Pascanu, R. (2017). Distral: Robust multitask reinforcement learning. In Advances in Neural Information Processing Systems (pp. 4496-4506).

---

> ### Author Response · Authors · 2018-11-26
> **Response to Reviewer 2**
>
> Response to Comparison to Other Algorithms, Ablation Study, More Test: In the revised paper, we did more insightful ablation study considering the various aspects of the proposed method. Please see Section 4.3 of the revised paper. There, we considered 4 parallel schemes based on TD3: 1) distributed RL TD3, 2) Experience-sharing-only TD3, 3) Reloading TD3, and 4) IPE-TD3, and the single-learner TD3. The 4 parallel schemes used the same number of learners. Since the x-axis of Fig. 4 is the environmental steps, the learner steps for the 4 parallel schemes can be obtained simply dividing the environmental steps by four. Only for the single-learner TD3 the environmental steps are the same as the learner steps. During the revision period, we also did more numerical evaluation of the proposed method applied to the SAC algorithms. The new result is shown in Appendices C and D
>
> Response to More Challenging Task: During the revision, following the reviewer’s comment, we considered a more challenging task, Humanoid, for which TD3 does not learn properly (this is shown in the original SAC paper) but SAC works. So, we applied IPE to SAC and obtained some preliminary result (not full result due to time limitation). The result is shown in Appendix D of the revised paper. The preliminary result shows that the proposed IPE method seems effective in the task of Humanoid.
>
> Response to More References and Previous Works: During the revision we searched related previous works, framed our work in the context of the previous work. We included many relevant references in Section 2 of the revised paper and explained the difference of the proposed method from the existing methods.
>
> Response to Terminology: By distance, we mean the distance between (parameterized) policies. We changed phi^i=> pi_{phi^i} in Figure 2. Since the policies are parameterized, the distance between policy with parameter 1 and policy with parameter 2 is related to the distance between policy parameter 1 and policy parameter 2. In the parameterized case, policy search is policy parameter search. Eq. (8) of the original paper (= eq. (4) of the revised paper) is explained as follows. Since TD3 and DDPG use deterministic policies, the policy pi(s) is a deterministic function of state s, and pi(s) for a given s is an action. However, this ||pi(s)-tilde{pi}(s)||^2 is averaged over s, as seen in eq. (7) of the original paper (or eq. (3) of the revised paper). So, the state-averaged distance is a distance between two functions pi(s) and tilde{pi}(s). For SAC, DQN, we used different distance measures. Please see Appendices C and E of the revised paper. We think that the specific choice of the definition of distance does not matter much if they are properly scaled. This is because any norm is equivalent, i.e., mutually bound each other. So, with proper scaling, different distance measures should work for the method.

---

### Author Response · Authors · 2018-11-26
**General Response to All Reviewers**

The authors of this paper thank the reviewers for their valuable comments to improve the paper. During the revision, we did more ablation study and more numerical evaluation of the proposed method applied to other algorithms such as SAC. We observed that the proposed method is effective in these settings too. As indicated by Reviewer 2, the new aspects of the proposed method are 1) to use a shared experience replay buffer for multiple learners and 2) to construct the augmented loss function for policy update instead of copying the best learner’s policy parameter to other learners.

-An ablation study on the various aspects of the proposed method is provided in Section 4.3 in the revised paper. It shows the effectiveness of the proposed way of fusing the best policy information.

-The proposed IPE is applied to SAC and the constructed IPE-SAC algorithm is applied to more difficult Humanoid (rllab). The result shows the promising result of the proposed method. See Appendices C and D.

-Reviewer 1 mentioned the motivation for parallel learning in robotics-inspired continuous-action RL. Even for such robotics systems, initial fast prototyping based on computer simulation becomes more and more important. In recent real-world control problems like autonomous cars, simulation-based initial learning is important. We cannot put an untrained unmanned car on the road or even on a test field to train the car from the scratch. This would injure people or destroy the car. So, in car companies, they first construct simulation systems for initial training. This can be done because Newtonian dynamics can be simulated and the control input is the handling wheel angle, brake, gas pedal, and gear ratio, which are all continuous. There are commercial companies providing such Newtonian-dynamics simulation platforms for fast initial computer-based learning, e.g. https://cyberbotics.com/ . They provide simulation platforms for development of complex robotics systems, autonomous cars, etc.. Thus, even for continuous-action controls, simulation-based learning is important. The trend is that people want to reduce the difficulty in tuning for real systems that Reviewer 1 mentioned by initial precise simulation-based learning. In this case, it is important to have an initial policy with good performance before actual road or field test, and the context of the paper makes sense.

Furthermore, the proposed IPE is general and can be applied to DQN with discrete actions. We constructed IPE-DQN. The pseudo-code of IPE-DQN is shown and how to construct IPE-DQN is explained in Appendix E of the revised paper. We tried simulation on several Atari games, but the simulation result could not be obtained within the short revision time.

-We have newly included many relevant references in Section 2 of the revised paper and explained the difference of the proposed method from the existing methods.

-During the revision, we slightly modified the update formula for beta to incorporate the speed of change in the parameter of the best learner. See eqs. 2 and 3 of the revised paper. Since we did not have enough time for tuning the parameters d_search, M, rho for the new beta update formula, we just used the parameters used for the original beta update formula. So, there may be some further gain after parameter tuning.

-During the revision, we change the performance metric of the parallel scheme to the maximum of the N learners not the average of the N learners, considering the goal of parallel learning.

-Although it is beyond the scope of the current paper, it is an interesting future work to combine the soft-way of using the best parameter information with hyperparameter tuning in PBT. In PBT, the parameter and hyperparameter of the best learner are copied to other learners, and the hyperpameter is slightly perturbed and the parameter is updated by SGD. Here, we can apply the idea of this paper. First, we can try the shared experience replay to collect diverse experiences from diverse learners for the same environment. This may or may not affect the performance of PBT. Second, we can try the soft-way of fusing the best learner information in PBT. We can measure the policy distance. So, if some bad policies are really far away from the best policy point, we just copy. On the other hand, if some non-best policies are around the distance d_search, we can apply the soft-fusing method to these nearby policies, while hyperparameter perturbation is on-going. This may fasten PBT for hyperparameter tuning.

We believe that the paper is worth acceptance and believe that the idea of the paper should be known widely to the research community so that researchers extensively try various possible ideas disseminating from the basic idea in the paper. For example, for each learner we can stimulate exploration and fast divergence from the best point by adding an entropy objective, multi-tier search radii, etc.

---

### Meta-Review · Area_Chair1 · 2018-12-14
**Interesting direction, still significant concerns on positioning with respect to wider literature and significance of contribution**

**Confidence:** 4
**Recommendation:** Reject

**Metareview:**

The authors present a new method for leveraging multiple parallel agents to speed RL in continuous action spaces. By monitoring the best performers, that information can be shared in a soft way to speed policy search. The problem space is interesting and faster learning is important. However, multiple reviewers [R2, R1] had significant concerns with how the work is framed with respect to the wider literature (even after the revisions), and some concerns over the significance of the performance improvements which seem primarily to come from early boosts. There is also additional related work on concurrent RL (Guo and Brunskill 2015; Dimakopoulou, Van Roy 2018 ; Dimakopoulou, Osband, Van Roy 2018) which provides some more formal considerations of the setting the authors consider, which would be good to reference.